# Process Evaluation of the Child and Youth Healthcare Intervention ‘Medical Advice for Sick-Reported Students in Primary School’ (MASS-PS)

**DOI:** 10.3390/ijerph19074409

**Published:** 2022-04-06

**Authors:** Esther Karen Pijl, Yvonne T. M. Vanneste, Jolanda J. P. Mathijssen, Frans J. M. Feron, Angelique E. de Rijk

**Affiliations:** 1Child and Youth Healthcare Department, GGD West-Brabant, 4816 CZ Breda, The Netherlands; 2Department of Social Medicine, Faculty of Health Medicine and Life Sciences, Care and Public Health Research Institute (CAPHRI), Maastricht University, 6200 MD Maastricht, The Netherlands; f.feron@maastrichtuniversity.nl (F.J.M.F.); angelique.derijk@maastrichtuniversity.nl (A.E.d.R.); 3Dutch Knowledge Centre for Youth Health (NCJ), 3527 GV Utrecht, The Netherlands; yvanneste@ncj.nl; 4Academic Collaborative Centre Youth, Tranzo, Tilburg University, 5000 LE Tilburg, The Netherlands; j.j.p.mathijssen@tilburguniversity.edu

**Keywords:** sickness absence, primary school, MASS, school absenteeism, child and youth healthcare, medical advice, implementation, process evaluation

## Abstract

School attendance is crucial for the development of a child. Sickness absence is the most common type of absenteeism and can be a red flag for underlying problems. To address sickness absence, the intervention Medical Advice for Sick-reported Students for Primary School (MASS-PS) was recently developed. It targets children at risk and is a school-based child and youth health care intervention. The present study is a process evaluation of the intervention. MASS-PS was implemented and evaluated in 29 schools in the West-Brabant region of the Netherlands, during three school years (2017–2020). Attendance coordinators (ACs) from the different schools were interviewed in six focus group interviews as well as in over 200 individual conversations, of which logbooks were kept. Content analysis was used based on a framework of implementation elements. During the first year of the study, the uptake was low. Changes were made by the project group to improve the uptake. The ACs generally considered the MASS-PS as compatible and relevant, but suggested improvements by adding a medical consultation function with a child and youth healthcare physician and increasing the threshold for selecting children at risk. They saw several personal benefits, although time was necessary to learn to use the intervention. An organisational barrier was the lack of teaching staff. A strength in the organisational structure was the appointment of ACs. A major event in the sociological structure was the COVID-19 pandemic. ACs felt that the intervention helped them keep track of sickness absence during the pandemic. The Medical Advice for Sick-reported Students for Primary School intervention was implemented successfully, and the process evaluation gave insight into possible improvements.

## 1. Introduction

Sickness absence among primary school pupils is a public health problem. School attendance is the foundation for learning and developing educational, social, and health-related skills [1,2,3,4]. Sickness absence is the most common reason for absenteeism and is explained by psychological, social, and health problems and can lead to lower educational achievement, as well as school drop-out [5,6,7].

For students in secondary education, an effective approach to address sickness absence was developed by a child and youth healthcare (CYH) organisation, in close cooperation with education providers [6]. This intervention, called ‘Medical Advice for Sick-reported Students’ (MASS), aims to reduce sickness absence and improve child well-being. It has recently been adapted to primary schools (PS) [8]. MASS-PS connects primary education and CYH services to identify and support children with extensive sickness absence (ESA). In MASS-PS, the parents, teacher and, if indicated, CYH physician discuss aspects of the pupil’s sickness absence and design and monitor a management plan to optimise health and maximise participation in school activities. The CYH physician gives medical advice from a biopsychosocial perspective in accordance with the age and cognitive and psychosocial development of the child [9]. An overview of key elements in MASS-PS is shown in Figure 1, a full description will be published elsewhere.

Child and youth health care services offer individual and community-based preventive healthcare in the Netherlands [9,10]. Based on the Public Health Act, CYH professionals promote and protect the physical and mental health of all children and monitor their development. In addition to this basic care, CYH offers specific interventions to reach children with an increased risk for health problems and reduced participation [11]. To identify children at risk, CYH services either reach out to parents and children directly or collaborate with other fields, such as education providers. The advantage of a school-based health intervention is the wide reach and frequent contact with children that schools offer [12,13]. However, it can be challenging to develop a feasible school-based health intervention, as the primary focus of teachers is on education, not health. It is important, therefore, to ensure user goals and intervention goals align, which can be improved by including user views in the design of the intervention [14,15].

The development of MASS specifically for primary school children was guided by intervention mapping [8]. Intervention mapping is a six-step process to structure the development of health-promoting interventions and incorporates stakeholders’ views to increase their usability and feasibility [16]. MASS-PS was shaped in steps 1–4, after which the intervention was implemented (step 5). Step 6 covers the process and effect evaluation of the intervention.

The aim of the present study is the process evaluation of MASS-PS with a focus on its implementation. Fleuren et al. [17] described elements that can affect implementation success at four levels (Table 1). These elements were used as a framework to analyse the implementation of MASS-PS in schools. This study aims to provide insight into the usability of MASS-PS and suggests possible improvements. Additionally, it aims to point at challenges while implementing a systematically planned, school-based, CYH intervention.

## 2. Materials and Methods

### 2.1. Setting

MASS-PS was developed in the period from January 2017 to August 2017 and pre-tested for feasibility among stakeholders in September 2017 [8] The evaluation was carried out from September 2017 to August 2020.

The present study was part of a larger research project exploring sickness absence in primary education. For that research project, 23 out of 265 primary schools in the region of West-Brabant in the Netherlands were selected for participation, 16 mainstream primary schools were selected at random and all seven special schools for primary education (SSPE) were approached. SSPEs are schools for children with mild learning difficulties, behavioural problems, and parenting problems, but they are not classified as special needs education in the Netherlands. Special needs education schools were excluded because the organisational structure is very different from regular education. Of the 23 selected schools, 10 mainstream primary schools and six SSPE agreed to participate and implement MASS-PS.

Since the uptake of the intervention in 2017/2018 was low, more schools were recruited in September 2018 by sending an invitation email to all schools in the region. At this stage, a further 14 schools agreed to participate in the study. The participating schools had between 64 and 495 pupils (median 210). Out of the 29 schools, 13 were located in an urban environment. The participating schools appointed an attendance coordinator (AC) to co-ordinate the implementation and use of MASS-PS.

### 2.2. Reach of MASS-PS

To determine the number of children reached by the intervention, the ACs provided data for the school years 2018–2019 and 2019–2020 on the number of children identified and referred to external experts in the context of MASS-PS. Because of the low uptake of the intervention, the ACs of the original 15 schools did not supply data for 2017–2018.

### 2.3. Interviews and Logbooks

Both focus group interview data and logbook data were collected. This method allowed for data triangulation between data obtained in a setting led by the researcher and those from a setting led by the users. To gain insight into the usability of MASS-PS, focus group interviews were held with the ACs of the participating schools, enabling them to share their own experiences and react to the experiences in other schools. Each school year (2017–2018, 2018–2019, and 2019–2020), two group interviews were held, with three to six participants in order to achieve data saturation. All ACs were invited to the focus groups and joined based on availability, resulting in a different grouping every year. The interviews were conducted by the first researcher and a research assistant and lasted an hour on average. The first four interviews were held face-to-face, while the last two were online due to restrictions from the COVID-19 pandemic. Topic guides for the interviews focussed on the adopters’ views on MASS-PS, whether the intervention was used as planned, and what factors enabled or impeded the intended use. The interviews were recorded and transcribed verbatim.

In addition to the focus group interviews, researcher E.P. maintained logs when she visited the schools every one to three months in 2018–2019 and 2019–2020 to discuss the use of MASS-PS with the AC—an action originally decided upon by the project group to improve the implementation. Logs held notes from more than 200 conversations. While the interviews were led by researcher E.P., based on topic guides, the individual meetings were led by the individual ACs, while E.P. made notes, the topics were based on the ACs’ recent experiences and events.

### 2.4. Analysis

The focus group interviews and logs were analysed through content analysis using the elements described in Table 1 as a framework. Comments were coded based on each of the elements by two researchers. In case of disagreement, all four authors discussed the coding until consensus was reached. All research team members read all data and participated in discussion of the results at several points in time. The research team members varied regarding their expertise in child and youth healthcare, school absenteeism, return to work, psychology social work, and epidemiology.

## 3. Results

MASS-PS was used by the participating schools between September 2017 and August 2020. In the first school year, uptake of the intervention was low. Therefore, the research team decided on new actions to support implementation, based on the first focus group interviews and a stakeholder advisory group. Identification software was created, the threshold for extensive sickness absence was adjusted, and regular consultations between author EP and each AC were held monthly. These consultations were designed to improve the organisation of MASS-PS in schools, but it quickly became clear that the ACs felt a need for medical advice, which E.P., as a CYH physician, could also provide. The logs of these regular consultation meetings, as well as the focus group interviews, resulted in a large amount of data on the implementation of MASS-PS and on the reach of the intervention during the research period.

### 3.1. Participation and Reach

Of the 29 schools that applied for participation, 20 participated during the entire research period. While using MASS-PS, they identified 1220 pupils with extensive sickness absence in 2018–2020, spoke to 489 parents (40%) about the pupils’ sickness absence and referred 136 pupils to external experts.

At five schools, participation was stopped during the study period. Three of the ACs at these schools stated that the research activities, rather than the intervention, took too much time. At another school, a change of principal was mentioned, and at one school a major reorganisation was reported as the main reason to end participation. Another four schools never got to implement the intervention as a direct result of a change of principal or AC. Of the participating schools, 11 out of 20 also had a change of principal and/or AC.

The views of the ACs of the 20 participating schools were used for the process evaluation. Additionally, for the five schools where participation ended early, the ACs’ views were used that had been collected before participation ended. There were no noticeable differences in the focus group discussions or individual conversations between the schools that were selected in 2017, compared to the schools that were recruited in 2018.

### 3.2. Experiences with MASS-PS

The information collected in the logbooks and focus group interviews were categorised based on the elements associated with the intervention, the user, organisation, and political context and legislation. Quotes from the ACs are reproduced in Table 2.

#### 3.2.1. Elements Associated with the Intervention

*Procedural clarity*—The intervention was generally considered to be clear.

*Correctness*—The main issue with the correctness of the intervention was the threshold for ESA, which was initially set as more than 6 days or more than three periods of sickness absence. During the focus groups held in the first year, ACs shared that over 80% of pupils in some younger classes fit these criteria, defeating the purpose of selecting those most at risk. This demotivated the ACs and teachers, and they reverted back to subjectively selecting those at risk. In 2018, the criteria were changed by the researchers to more than 9 days or more than four periods of sickness absence, which was happily accepted by the ACs.

Some ACs were unsure of how to handle four- and five-year-olds. They suggested that there is more absenteeism because children have to get used to going to school and due to infectious diseases. At four years old, school attendance is not mandatory in the Netherlands, and the ACs suggested that both school personnel and parents consider school attendance to be less important than in later years. All those factors combined to create confusion. One school supplied a solution in a focus group that was well accepted by many other ACs: to consider the process of getting used to school as an integration program, not as sickness absence.

*Completeness*—Almost all ACs were clear that the intervention needs to include a consultation option with a CYH professional. After children with ESA had been identified, the ACs wanted medical advice from a CYH physician to help them choose which children needed additional help and which external experts should be involved. This option was added to the intervention in 2018 and was universally used by ACs. Nearly every logbook entry included notes on medical advice for individual children.

*Complexity*—When a child is absent from school, the teachers could find it difficult to know how to report the absenteeism, according to the ACs. They felt a need to standardise reasons for absenteeism further, for example when a child is going to the hospital: is it sickness absence or a doctor’s visit?

ACs revealed that some teachers found the conversation with parents easy because they already have a good relationship with them. Other teachers found it very difficult, and the AC would then have more work encouraging the teacher to have the conversation and supporting them during the conversation. Many ACs considered conversations with parents to be easier because of MASS-PS, as it provided an objective conversation starter: absenteeism numbers.

*Compatibility*—Generally, the steps of MASS-PS were considered to fit well in the day-to-day work of the schools. However, one reported problem with compatibility was the multidisciplinary team, which only worked if regular meetings were already part of the school structure. Schools without multidisciplinary teams were unable to organise these meetings and, thus, skipped this step. Instead, the AC would choose whether to involve the CYH physician or another external expert, often using the above-mentioned consultation function first.

Other incompatibilities were only mentioned by one or a few ACs, including that the criteria were considered too strict during a flu outbreak, when many children were reported as sick. Additionally, some schools had a lower prevalence of ESA and noticed that no new children would be identified if they checked every month, so they checked every 6 to 8 weeks instead.

*Observability*—Many ACs recognised improvements in the recording of absenteeism and gained insight into absenteeism patterns in their schools. Many also noticed changes in the prevalence of sickness absence and school personnel’s and parents’ attitude towards sickness absence. They identified children with ESA and underlying problems that would not have been noticed otherwise.

*Relevance for the client*—During the focus groups, the ACs reported that communication with the parents had improved with MASS-PS and that earlier action led to easier solutions for the child.

ACs discussed that both school personnel and parents consider school attendance to be less important when a child is young. School absenteeism was thus not always seen as a problem. Several ACs noticed that increasing personal contact about the way absenteeism is reported helped to reduce sickness absence.

#### 3.2.2. Elements Associated with the User

*Personal benefits and drawbacks*—ACs found that working with MASS-PS increased their work pleasure, due to greater awareness and insight into sickness absence and because they knew what to do with sick-reported pupils. Some happily reported that communication with parents was easier for them, and they noticed a decrease in sickness absence rates.

The main downside they mentioned is that using MASS-PS properly takes time. Every now and then, an AC would half-jokingly say that it can be nice for the teacher and class when a specific child is absent, suggesting that ending absenteeism is not always a benefit in the short term.

*Outcome expectations*—Only minor remarks were made on this topic, such as one mention that a good school climate, or discussing absenteeism with parents, seems to reduce absenteeism.

*Professional obligation*—According to several ACs, teachers focus on teaching and do not consider sickness absence as their responsibility.

*Client satisfaction*—Some positive experiences were reported where parents were happy with the attention.

*Client cooperation*—A few ACs believed that some parents report their child as sick far too easily, due to a lack of awareness of sickness absence as a problem, especially for four- and five-year-olds.

ACs also expect that some parents will not want to work with external experts, due to negative experiences with the experts’ organisations. On the other hand, ACs also mentioned that involvement of the external experts could help parents realise the importance of school attendance.

ACs noticed that improving cooperation with parents takes time. They considered the attitude of care rather than control as helpful, as well as talking about sickness absence and showing parents a visual of the absenteeism.

*Social support*—One of the biggest challenges the ACs faced was getting all teachers to use MASS-PS. It took time to implement, especially in the larger schools. There were some examples of a lack of support among professionals, and some examples of great support.

*Subjective and descriptive norms*—Norms were not mentioned by the ACs.

*Self-efficacy*—Once familiar with MASS-PS, ACs felt able to use it and reported that teachers were getting more confident too. The intervention made the ACs feel more secure when addressing sickness absence.

*Knowledge*—Only minor remarks were made, for example that ACs used team meetings to inform other school personnel.

*Awareness of content*—Awareness of the content of, and the need for, MASS-PS was deemed crucial in the interviews, and a clear progression was seen from 2018 to 2020. Awareness among teachers and parents grew, though not in all schools.

#### 3.2.3. Elements Associated with the Organisation

*Formal arrangements*—Hardly any school had formal arrangements during the research period. The ACs believed it would be necessary to put policy in place for the continued use of MASS-PS. For example, they found it crucial to plan the identification of ESA, otherwise it would only happen when a meeting with the researcher was scheduled. Some ideas for improvement were shared, such as adding absenteeism as a standard topic in parent–teacher meetings or on report cards.

*Replacement when staff leave*—Changes in the school teams could hinder the use of MASS-PS. Finding new staff was a challenge in some schools and, when found, new staff needed to be trained in the use of MASS-PS.

*Staff capacity*—Some ACs reported a lack of staff capacity.

*Financial resources*—Financial resources were not mentioned.

*Time available*—Setting aside time to identify ESA was one of the biggest challenges for the ACs. Even though they claimed to see a major added value and wanted to do it, they did not find the time because they had so many other activities. MASS-PS was not in the forefront of their minds, and often only the regular meetings with the researcher prompted action. Near the end of the research period, more and more ACs did start to make time for identification, which they managed through careful planning.

*Material resources and facilities*—It became clear in the first year that identifying ESA was not possible in the current school software and, thus, it took far too much time, according to the ACs. Therefore, software was designed specifically for MASS-PS that uses data from the school registration systems to identify ESA. Downsides of this change were that this new software had a learning curve and needed technical support to keep working. However, the experienced ACs considered it a great addition as it gave them more options and insight into all absenteeism. All agreed that it would be even better if it could be integrated into the school registration systems.

*Coordinator*—ACs rarely spoke about their own role directly. Their importance shines through other remarks made, such as how they have to encourage teachers to act, how they support teachers in their conversation with parents, and how they ensure that the registration and identification of children with ESA happens.

*Unsettled organisation*—There was one case of reorganisation which may have hampered implementation.

*Information accessible about use of innovation*—Only minor remarks were made.

*Performance feedback*—Performance feedback was not mentioned by the ACs.

#### 3.2.4. Elements Associated with the Socio-Political Context

*Legislation and regulations*—Some ACs were worried about the introduction of the EU’s general data protection regulation (GDPR) in 2018. The GDPR limits information sharing, which could make it more difficult to cooperate with CYH professionals and other external experts. However, the ACs found that information can be shared in different ways, in the interest of the child’s well-being.

*Pandemic*—As the research took place during the outbreak of the COVID-19 pandemic, this topic was also frequently mentioned in 2020. ACs mentioned an increase in sickness absence during the pandemic in some schools and barely any change in others. Some ACs found that MASS-PS helped them discuss corona-related anxiety. ACs also noticed that, because of online lessons, children with serious medical problems were able to participate more than before.

## 4. Discussion

The implementation in schools of the newly designed CYH intervention, MASS-PS, was evaluated through qualitative research with six focus group interviews and logbooks of over 200 conversations with ACs.

This process evaluation revealed a generally good implementation among the participating schools, particularly after the first implementation year. The final reach of identifying 1220 pupils with ESA and talking to almost 500 parents about ESA shows that the intervention was implemented and that the ACs had enough experience with the intervention to discuss its usability. The success of MASS-PS could be understood as being driven by elements on all four levels of implementation (intervention, user, organisation, and socio-political context) [17]. The perceived positive effect of MASS-PS on children’s well-being especially appealed to both ACs and teachers and motivated its continued use. This showed the importance of alignment between user goals and intervention. Barriers were mainly found at the organisational level: participation ended quite frequently because of a change in a key figure (AC or principal), even if it appeared possible to continue if the key figure supported MASS-PS and was able to pass the role on.

Focussing on the elements associated with the *intervention*, an obstacle in the first year was that the threshold for extensive sickness absence was regarded as far too low by the ACs. A low threshold defeats the purpose of specifically selecting children at risk, and the selected group will be too large to manage. This demotivates the user as the effect of the intervention is less visible. While identifying problems at an early stage is important for prevention in CYH, selecting children too early hinders implementation, especially in a school setting. Therefore, the researchers raised the threshold, and the new threshold of more than 9 days or more than four periods was happily accepted by the ACs.

For MASS-PS, the medical consultation function with a CYH physician was found to be crucial. While it was originally added by the researchers to improve the organisational side of implementation, it became clear there was a strong need among the ACs for medical advice on what they could do for individual children. The consultation function reinforces the use of MASS-PS and allows schools to have easy access to medical expertise, strengthening mutual understanding and collaboration.

MASS-PS provided teachers with the tools to talk to parents about sickness absence, such as a care perspective rather than a control perspective, and with the objectiveness of absenteeism numbers to start a conversation. Even so, ACs confirmed that talking to parents was regarded as difficult by many teachers, particularly concerning health issues. These difficulties might be related to a larger problem, as research on family involvement in schools suggests the need for improving parent–teacher communication, possibly through changes in teacher training programmes [18,19].

The improved absenteeism registration and the overview of pupils with ESA helped to motivate users. Interestingly, according to ACs, both school personnel and parents seemed to be less worried about ESA when a child is less than six years old, because school is considered to be less important at that age. In contrast, international research and policy makers stress the importance of early childhood education as the basis for success in life [20]. While increased absenteeism due to adjusting to school life or infectious diseases may be expected at that young age, ESA should not be tolerated as it might be a red flag for underlying problems that had not been noticed before the child attended school [21]. Awareness of the importance of school attendance in early childhood should be increased among both parents and teachers.

At the level of the *user*, the awareness of a health problem and its link to education was crucial for motivation, especially as teaching is the main focus for school personnel. Teaching does require a caring approach and attention to the child’s well-being. With the growing awareness of a link between school absenteeism and well-being, school personnel were more motivated to address absenteeism. MASS-PS includes both a collective approach, through the registration of all absenteeism and a threshold for ESA, and an individual approach when a child is identified to be at risk. This allows ACs and teachers the flexibility to tailor their approach to the child based on their own expertise, supported by a CYH physician or other experts when necessary. The ACs reported time constraints. It takes time away from other educational activities. Furthermore, it takes time to master the intervention and disseminate it among other school personnel. The dissemination is a well-known process and can take many years; the research period may be too short to see the full effect of implementing MASS-PS [22]. However, once they started working with MASS-PS, the ACs experienced enough benefits—in the insight into absenteeism rates, the improved contact with parents through care rather than control, and a decrease in absenteeism—to offset the downside of spending time. Moreover, glimpses of ACs internalising the MASS-PS method were seen, as more and more of them performed the identification step, and contacted CYH physicians, without being prompted.

The evaluation at the level of the *organisation* highlighted prerequisites that need to be met in order to successfully implement MASS-PS. The availability of an absenteeism registration program that allows for clear registration and the identification of extensive sickness absence is paramount. Especially because professionals from another field (teachers) have to select a group at risk for medical absence, selection should be easy and quick.

Other prerequisites for MASS-PS are the availability of an AC as a key figure, and sufficient staff capacity, replacement, and transfer of duties in case staff leaves. In the Netherlands and across Europe, there is a shortage of teachers, and the workload for school personnel is higher than nearly any other profession [23,24,25]. Thus, it is not only an organisational problem, it also has links to the socio-political context as teacher shortages and lack of political priority can make school-based health interventions unfeasible. It should be noted though, that MASS-PS in itself can improve efficiency with its focus on targeting and prevention of deterioration of biological, psychological, and social problems.

Examining the *socio-political context* further, the biggest event during this research period was the start of the COVID-19 pandemic, which increased the prevalence of sickness absence and caused a school lockdown for 2 months. Thus, attention to school absenteeism was even more important as research has shown that the pandemic increased school absenteeism, missed lessons, and differences between pupils [26]. MASS-PS was still used during this period and was reported to have helped in a few schools, showing the compatibility of MASS-PS, the need for such an intervention to tackle absenteeism, and its efficiency.

In the present study, the framework for elements of implementation was not used during data collection to minimise information bias. The framework allowed for structing the findings. However, it did not highlight the weight of each of the factors compared to each other and, while most of the elements could be found in the data, it is not known if elements not described by the framework were missed. Additionally, while data saturation was reached on each level of implementation, not all specific elements were mentioned by the ACs, such as performance feedback, descriptive and subjective norms, and financial resources. It is not known whether the ACs experienced any problems or benefits or simply had little knowledge of these elements. For example, some ACs may not have a financial role in their school.

### 4.1. Methodological Strengths and Limitations

Various factors contributed to the study’s trustworthiness, defined as the credibility, dependability, transferability, and confirmability of the data analysis [27]. A strength of this study is the rich data collected, with both in-depth focus group interviews and logbook information from more than 200 conversations with individual ACs. This data triangulation added to the trustworthiness, specifically the study’s credibility, as the data often showed similar sentiments in both the focus group interviews and logbooks. The data were mainly collected by one researcher which may have added to a confidential atmosphere that stimulated the ACs to be more honest. To reduce bias and improve dependability of the method, this researcher had frequent reflective meetings with all authors during data collection. Finally, all data were read by the research team, the data were analysed by two different researchers, and the findings discussed among all authors at different moments.

This study found that the implementation of MASS-PS in primary schools was low during the first year, and no data on reach and implementation were shared by ACs that year. To improve implementation, three measures were taken at the level of the intervention itself and the organisation: adjustment of the threshold for ESA, adding a consultation function with regular meetings, and creating software for the identification of ESA. Additionally, more schools were included to increase data collection in following years. These measures had a direct impact on the finding, for example, in the first year ACs reported issues with implementing the threshold criteria and identifying extensive sickness absence. In order to be able to implement further steps of the intervention, it was necessary to improve the criteria, otherwise it would not have been possible to study the implementation of the full intervention. Problems with the criteria were not reported after 2018. The changes led to a better uptake, showing that evaluation and adjustments are crucial during the implementation of an intervention. The addition of regular meetings and more participating schools led to more data and data saturation, as mentioned above.

While the findings are theoretically transferable to all schools—particularly schools motivated to tackle sickness absence—selection bias is rather probable, as the participating schools may have been more motivated to tackle sickness absence than the schools that declined participation. The schools approached in 2017 were selected at random to minimise this bias. MASS-PS can probably be implemented successfully in other schools in the Netherlands, as the difference between schools are relatively small at all levels of implementation. The comprehensive description of the results, which has been discussed on several occasions with all authors and included illustrating quotes and logs, contributes to confirmability of the study. Whether MASS-PS can be implemented well in schools outside of the Netherlands needs to be studied, as the socio-political context or organisational structures may differ. Moreover, this process evaluation does not demonstrate the effectiveness of MASS-PS. The effect evaluation requires an intervention to be successfully implemented first. The findings that ACs believed there to be a positive effect and that teachers talked to parents about absenteeism in 489 cases of the initial 1220 cases at risk (40%) might indicate an effect, but this needs to be substantiated in a planned effect evaluation.

### 4.2. Recommendation for Further Research

The MASS-PS intervention targets primary school pupils as a whole, but the ACs suggested that both parents and teachers believe that sickness absence has less of an impact on four- and five-year-olds than older children. Future research should examine the longitudinal effects of the intervention in younger children separately.

### 4.3. Recommendations for MASS-PS

This study provided several options that could improve MASS-PS and its implementation.

#### 4.3.1. Changes That Could Improve MASS-PS

-Adjusting the threshold of more than three periods or more than 6 days in a school year to more than four periods or more than 9 days.-Adding a consultation function by a CYH professional to reinforce MASS-PS on an organisational level and to give medical advice on actions to take for individual pupils.

#### 4.3.2. Prerequisites That Could Improve the Implementation

-Supplying software to identify ESA to all schools.-Unifying the recording of the reason for absenteeism (e.g., sickness absence, doctor’s visits, other authorised absence, tardiness, and other unauthorised absence).-Making the multi-disciplinary team part of the school structure.-Including the identification of ESA as a standing item on the school agenda and an official task in the school organisation.

### 4.4. Recommendations for the Implementation of School-Based Health Iinterventions

When designing and implementing a school-based health intervention focussed on medical advice for sickness absence that targets an at-risk group, it is crucial to ensure awareness of both the health problem and the benefits for schools in addressing it. In the present study, ACs reported that the intervention was easier to use as soon as teachers and parents became aware of the impact of sickness absence on well-being.

Additionally, the remarks on the difficulty teachers had in talking to parents suggested that it could be important to keep the capabilities of school personnel in mind and include either support or training in a school-based health intervention.

Finally, sufficient teaching staff is crucial for the execution of a school-based health intervention such as MASS-PS. The extra responsibilities should not lead to overload and, consequently, sickness absence among teachers. However, this is a real possibility due to international teacher shortages and high work stress. However, this can only be achieved by political action, which is beyond the scope of this study. Furthermore, the timing of introducing MASS-PS should be aligned with sufficient available staff.

## 5. Conclusions

The newly designed school-based child and youth healthcare intervention MASS-PS was implemented and the process evaluated. Elements associated with all levels of implementation contributed to the usability of MASS-PS, especially the alignment between the goal of the intervention and of the users to improve the child’s well-being. Emphasizing the benefits for education was crucial for implementation. A major barrier for the implementation of MASS-PS and—in a similar vein—other school-based health interventions, is shortage of staff, which requires actions at the socio-political level. Even so, MASS-PS supports efficient absence management as it targets those pupils most in need and prevents further deterioration of underlying problems. The present study found that MASS-PS can be improved by the addition of an adequate threshold for ESA and by supporting school personnel with the option of a medical consultation for the child and his or her caregivers with a child and youth healthcare physician.

## Figures and Tables

**Figure 1 ijerph-19-04409-f001:**
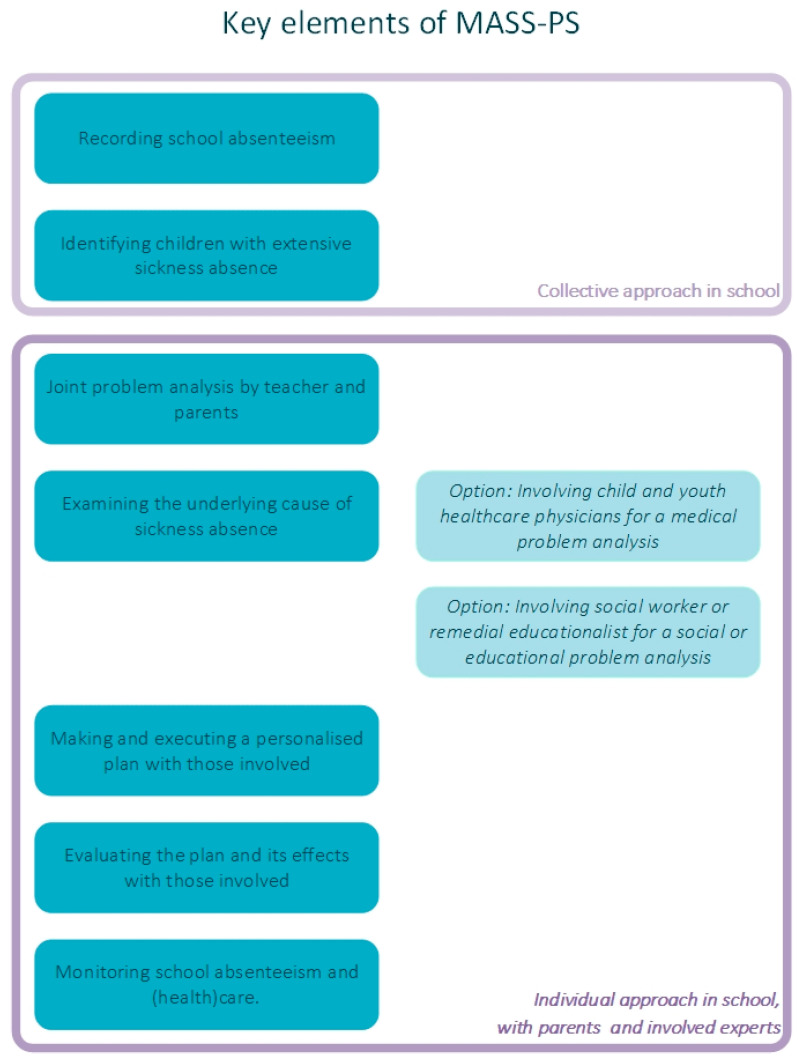
Key elements of the MASS-PS intervention. MASS-PS: Medical advice for sick-reported students in primary school.

**Table 1 ijerph-19-04409-t001:** Overview of the elements of implementation.

**Elements Associated with the Innovation**	
Procedural clarity	Compatibility
Correctness	Observability
Completeness	Relevance for the client
Complexity	
**Elements associated with the adopting person (user)**	
Personal benefits/drawbacks	Descriptive norm
Outcome expectations	Subjective norm
Professional obligation	Self-efficacy
Client/patient cooperation	Awareness of content of innovation
Social support	
**Elements associated with the organisation**	
Formal arrangements by management	Material resources and facilities
Replacement when staff leave	Coordinator
Staff capacity	Unsettled organisation
Financial resources	Information accessible about use of innovation
Time available	Performance feedback
**Elements associated with the socio-political context**	
Legislation and regulations	

**Table 2 ijerph-19-04409-t002:** Quotes from attendance coordinators sorted by the elements of implementation.

Theme	Element	Quotes from Absenteeism Coordinators
**The innovation**	Procedural clarity	*“This system distinctly shows how far everyone’s responsibility goes. The collaboration is clear-cut.” —2019*
	Correctness	*“We wonder if the criteria are too strict? (…) Actually, we know who the children with problems are. It’s still good to do, although I think we are lucky in our village as problems aren’t that big here.”—2018*
	Completeness	*“I always do it with you [meaning researcher & child and youth health care physician EP], we discuss the identified children and then get the ball rolling.”—2020*
	Complexity	*“One teacher categorises a dentist visit as a doctor’s visit, the other calls it other authorised absence, so we changed the options with [researcher] EP. Less choice in the registration system makes it easier to pick the right category.”—2020*
		*“I’ve noticed that our teachers find it difficult to talk to parents about absenteeism. I have to remind them (…) not to actually say ‘your child is absent far too often’.”—2019*
	Compatibility	*“For our school, identifying absenteeism every six to eight weeks is fitting. (…) With the possibility of more frequent checks when there are signs of increased absenteeism, but that’s up to the school to decide.”—2020*
	Observability	*“Teachers often think it’s authorised absenteeism, because they fill in that it’s this or that’, (…) When you ask parents, however, it may be confirmed, or it may be very different than what we thought. Knowing the problem makes it easy to tackle.”—2019*
	Relevance for the client	*“You tackle the problem earlier (…), before, a child had 40 periods of absence, but now, they’ve already been identified. That makes it easier to involve the teacher, and for the teacher to involve the parents. That’s an improvement.”*
		*“I believe that parents don’t recognise the necessity as of yet’* *‘Actually, I’ve noticed a difference between younger and older groups. (…) Parents are more focussed on absenteeism of their child in group 8, than in group 2’* *‘And the teacher does the same of course, saying ‘this can’t happen next year’.”—Two ACs 2019*
**The user**	Personal benefits/drawbacks	*“The whole trajectory (…) has given me insights, about talking to parents and even children. (…) I really appreciate that. It provides a kind of norm, a criterion that shows what is normal and what is excessive.”—2020*
		*“For me, it’s that you’re made aware so much earlier. The children are noticed so much sooner and you can start the conversation with parents in an earlier stage.”—2020*
		*“Sometimes, you’re kind of relieved that a few pupils are ill, it’s quiet.”—2018*
	Outcome expectations	*Minor remarks*
	Professional obligation	*“Teachers don’t see it as their job (…), they teach whoever is in the class. The absenteeism is the parents’ responsibility, the teacher is responsible for brushing up the knowledge once the child is back.”—Logbook*
	Client/patient satisfaction	*“When I talk to parents myself, I don’t get the impression that they mind. They seem to appreciate that absenteeism is identified and we care about it.”—2019*
	Client/patient cooperation	*“Sickness absence can sometimes seem an easy way to be able to go to a theme park (…), a five-year-old will blurt that out by the way.”—Logbook*
		*“As soon as we mention the CYHP or nurse, we see a drop in absenteeism. It makes parents think.”*
	Social support	*“What I do notice is some teachers are very enthusiastic and good at it. Some other teachers seem afraid that it will create more work for them, when in fact it doesn’t have to.”—2018*
		*“I am the only SNC on staff, so my communication with the teachers is easy (…). And I can see that they really care too. This was a team decision, because the teachers were also concerned about the children with high levels of absenteeism but we didn’t know exactly what to do (…), so MASS-PS is great.”—2019*
	Descriptive norm	*Not mentioned*
	Subjective norm	*Not mentioned*
	Self-efficacy	*“I notice that once a teacher had had a few conversations with parents [about absenteeism], they say ‘it’s easier than we thought’. Especially because you can use MASS, which gives peace of mind. You can hide behind the numbers and don’t have to jump to conclusions.”—2019*
	Awareness of content of innovation	*“Word of mouth around the village is that sickness absence is monitored now, and that has an effect on the absenteeism in school.”—Logbook*
**The organisation**	Formal ramification by management	*“When I look at myself, I have to ensure that I make time to monitor absenteeism. The issues of the day catch up to us all (…) My advice to starting schools would be: make it a recurring item on the agenda.”—2020*
	Replacement when staff leave	*“We need to keep explaining the program to teachers, (…) Because of changes in colleagues, or other changes in the teams, (…), it [the intervention] soon becomes less like the original.”—2020*
	Staff capacity	*“There are a few positions I can’t fill, my SNC is absent long-term for example and I’m missing an important link to use this intervention. (…) It changes too much, so I’ll try, but no guarantees.”—2018*
	Financial resources	*Not mentioned*
	Time available	*“It takes time, not that that’s bad or anything, time just has to be available.”—2018* *“We only do it when the researcher comes, that’s disappointing isn’t it?”—logbook*
	Material resources and facilities	*“It’s a pity that the functionalities of the MASS-PS program aren’t implemented in the regular school registration software.”—2020*
	Coordinator	*“We’ve had a study day in September and I am kind of the driving force behind this intervention in school, together with the principal and special needs coordinators.”—2018*
	Unsettled organisation	*“I’ll be honest, it’s still something I want to implement properly in my school. It just hasn’t happened yet because of all the turbulence and busy schedules.”—2018*
	Information accessible about use of innovation	*Minor remarks*
	Performance feedback	*Not mentioned*
**The socio-political context**	Legislation and regulations	*“What is the role of the GDPR? (…). We’re not sure yet what is allowed and what isn’t.”—2018*
	Pandemic	*“Children with, for example, a serious intestinal disorder, who couldn’t be in school before the pandemic, (…) can now log in to the class from home and miss fewer lessons than before.”—2020*
		*“Now that the school lockdown is over, the sickness absence is skyrocketing again. You can’t blame people because you have to stay home with any symptoms of a cold, but because of that we’ve lost our grip on sickness absence again.”—2020*
		*“After Corona, sickness absence was lower than before. We’ve had no negative effect on absenteeism. (…) Yes, the occasional pupil goes for a test, is absent for a while, but nothing comes out of it.”—2020*

## Data Availability

Data will not be made publicly available as it contains information that would compromise participant privacy. Data can be made available on request.

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
