# Peer review of "Process Evaluation of the Child and Youth Healthcare Intervention ‘Medical Advice for Sick-Reported Students in Primary School’ (MASS-PS)"

_ijerph, 2022, doi:10.3390/ijerph19074409_

Round 1

Reviewer 1 Report

Dear authors, thank you for allowing me to review this exciting paper assessing qualitatively the implementation of a medical advice intervention targeted at identifying sick-reported students in primary school.

I found this manuscript well-designed and methodologically sound, and focused on a relevant topic for public health sciences. Despite the paper being well-written, I found some minor comments that would, if addressed, improve the quality of the study.  

Figure 1: would you please improve the quality of this image? It would enhance its readability. 

Page 3, line 94: maybe "oversee" should be replaced with "check"

In the data analysis section, you should consider reporting how you managed the data saturation issue.

I found some aspects in the results section deserving better consideration in the discussion.

On page 4, you have described how the research team decided on new actions to support implementation. Could these actions have influenced your findings?

Along with all the results section, you emphasised the issue of time constraints related to implementing the MASS-PS that has not been discussed in the discussion.

In the discussion (page 9, lines 315-318), you have considered some issues related to staffing and lack of resources, suggesting recruiting more teachers as a possible solution (page 12, lines 460-462). This seems a trivial solution. Are there some other viable solutions that could be applied?

Page 11, lines 399-401 could be the choice you made of enlarging your sample have influenced your findings?

Moreover, in the limits/strengths section, you should provide information about the strategies you used to ensure trustworthiness. 

I hope my comments will help you improve the quality of this promising paper. 

Reviewer 2 Report

This article evaluates the process of implementing an intervention in the school environment to control absenteeism due to illness.
It is a very interesting and current article, especially in these times of pandemic and the presence of prevalent diseases in the child and adolescent population.
To improve the article the authors should consider the following changes.
The type of study they have carried out has been outlined in the discussion section, and it should not be there, but in material and method.
Regarding the selection strategy of the participating schools, there is a bias, which, although it is explained later, must make a greater effort to explain the influence that this selection has been able to obtain in various stages, since it can influence the results obtained. , if they are schools with more or better resources, they had previous information about the evolution of the project in the schools of the area.....

Along the same lines, there are modifications in certain aspects of the program as the project progresses, it should be very clear if these changes have had any influence on the results obtained, especially if there are differences in the results obtained in the centers that had certain criteria and in the centers that adhere to the project later with other different criteria.
The author must indicate whether data saturation was reached in the interviews conducted throughout the study.
The discussion must be improved, and compare it, discuss it, debate it with other studies.
Please review the bibliography and adapt it to the regulations of the journal.
